# Passionfruit (*Passiflora edulis*) Peel Powder Stimulates the Immune and Antioxidant Defense System in Nile Tilapia, *Oreochromis niloticus*, Cultivated in a Biofloc System

Piyatida Outama [1], Nguyen Vu Linh [1], Chinh Le Xuan [1], Supreya Wannavijit [1], Sudaporn Tongsiri [2], Chanagun Chitmanat [2], Napatsorn Montha [1] and Hien Van Doan [1,3,*]

1   Department of Animal and Aquatic Sciences, Faculty of Agriculture, Chiang Mai University, Chiang Mai 50200, Thailand
2   Faculty of Fisheries Technology and Aquatic Resources, Maejo University, Chiang Mai 50290, Thailand
3   Innovative Agriculture Research Center, Faculty of Agriculture, Chiang Mai University, Chiang Mai 50200, Thailand
*   Correspondence: hien.d@cmu.ac.th

**Abstract:** This study aimed to assess the impacts of dietary supplementation with passionfruit (*Passiflora edulis*) peel powder (PSPP) on the growth, immune response, and expression of immune and antioxidant-related genes in Nile tilapia (*Oreochromis niloticus*) maintained in a biofloc system. Fish were fed basal diets supplemented with different doses of PSPP at 10 g kg$^{-1}$ (PSPP10), 20 g kg$^{-1}$ (PSPP20), 40 g kg$^{-1}$ (PSPP40), and 80 g kg$^{-1}$ (PSPP80). The basal diet, without PSPP-supplementation, was used as a control at 0 g kg$^{-1}$ (PSPP0). We observed that the dietary supplementation groups fed different levels of PSPP exhibited no substantial difference or only slight increases in growth performance and immunological response in Nile tilapia ($p > 0.05$), whereas fish fed diets supplemented with PSPP at concentrations of 10 g kg$^{-1}$, 20 g kg$^{-1}$, and 40 g kg$^{-1}$ had significantly higher mRNA transcripts (approximately 1.5–4.5 fold) of immune (*il-1*, *il-8*, and *lbp*) and antioxidant (*gst-α*, *gpx*, and *gsr*) gene expressions than fish in the control treatment group (0 g kg$^{-1}$). These findings suggest that dietary supplementation with PSPP may effectively stimulate the immune and antioxidant defense system and may function as feed additives in Nile tilapia cultured in a biofloc system.

**Keywords:** feed additives; immune gene expressions; antioxidant defensive system

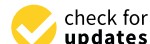



## 1. Introduction

Nile tilapia (*Oreochromis niloticus*) has been extensively produced in more than 100 nations globally, generating around 7.3 million tons in 2021, because of its flexibility, high growth, resistance to stress and disease, and great economic value [1–4]. Nonetheless, like with any intensively cultured fish, tilapia farming imposes significant strains on the water quality for fish farming and increases the occurrence of pathogenic infections—especially bacterial diseases [5,6]—resulting in a high mortality rate (up to 95%) and massive economic losses [7,8]. Antibiotics and chemotherapeutics have been commonly used by farmers all over the world to control disease outbreaks in fish farming [9]. However, these activities have caused the outgrowth of antibiotic-resistant bacteria [10,11]. Due to restrictions on the use of antibiotics in aquaculture, the development of innovative ways to supply appropriate feed additives and develop cost-effective methods of disease prevention and treatment for fish has become a top concern [12,13]. Consequently, natural immunostimulants (such as prebiotics, probiotics, and synbiotics) are promising alternatives for modifying the bacterial population and attempting to control infectious disease outbreaks in aquaculture by enhancing dietary intake, nutritional absorption, and immune defense systems in aquatic animals [14–17]. In this respect, fruit by-products have been identified as potential supplements in the diets of aquatic species [18–20]. Fruit by-products used as feed additives

have the potential to minimize waste, reduce aquafeed costs, and offer raw materials for the nutritional sectors [21]. Additionally, utilizing these by- and co-products would also have a positive influence on the environment and provide additional advantages to farmers [22,23].

Passionfruit (*Passiflora edulis*) is a species of the Passifloraceae family, which has more than 500 species [24]. It is found mostly in North America, but also in tropical and subtropical Southeast Asia, Australia, and New Zealand due to its economic and medicinal properties [25,26]. The passionfruit extract market is projected to reach USD 1028.6 million by 2029 [27]. Peels are created in great amounts during the processing of passionfruit to produce passionfruit juice [28]. Moreover, passionfruit peel is a by-product of the fruit processing industry that makes up around 50% of the weight of the fruit [29], which is typically thrown away as waste [30,31]. The passionfruit peel includes a variety of bioactive components, including phenolic compounds, flavonoids, cyanogenic chemicals, anthocyanin, minerals, polysaccharides, and vitamins [28,32–35]. Numerous investigations using passionfruit by-products as feed additives have been conducted on sheep [36], swine [37,38], quail [39,40], and poultry [41,42]. For fish farming, the incorporation of passionfruit seed meal (including its oil residue) in diets for tambaqui (*Colossoma macropomum*) [43,44] and passionfruit juice in tilapia has been investigated [45]. However, there have been few studies on the influences of passionfruit peel powder (PSPP) on the growth and overall wellbeing of common commercial fish species—particularly Nile tilapia.

Biofloc technology (BFT) is an alternative approach that mixes aggregates of algae, protozoa, or bacteria with particulate organic substances to improve water quality, waste treatment, and disease prevention in intensive aquaculture systems. It has been proposed as a cost-effective alternative to intensive systems since it improves water quality without requiring water exchange and produces microbial protein for aquatic species. Biofloc is a microbial community composed mostly of prokaryotic and eukaryotic microorganisms and different organic particulates [46–48]. Biofloc functions as a nutrition supply for aquatic creatures in this system, assisting in growth enhancement, pathogen reduction, and zero-water exchange. Additionally, BFT has a beneficial impact on the host immune system, increasing resistance to diseases and infections [49,50]. Therefore, this study aimed to assess the influence of dietary supplementation with powdered passionfruit peel on growth, immunological responses, and the expression of key immune-antioxidant-related genes in Nile tilapia raised in a biofloc system.

## 2. Materials and Methods

### 2.1. Preparation of Powdered Passionfruit Peel and Experimental Diets

Passionfruit was obtained from local markets at Chiang Mai (Thailand). Passionfruit peels were oven-dried at 60 °C for 48 h. The dried peel was then ground into a powder using a mill and stored at 4 °C until used in the fishes' diets. To prepare the dough, the feedstuffs were blended and then soybean oil and distilled water were added. The dough was then pelleted (2 mm in diameter) in a pelleting machine. Pellets were dried at 50 °C to attain 10% moisture and stored in sealed polyethylene bags at 4 °C until used. The proximate composition of the diets was determined using the techniques recommended by AOAC [51], and the crude fat content was measured using an extractable matter machine (Soxtec™ 8000, Hilleroed, Denmark). The basal diets were modified according to the descriptions reported previously [52], which proved their suitability for Nile tilapia by their different levels of PSPP. The ingredients and elemental composition of the basal diets are shown in Table 1.

**Table 1.** Formulation and proximate composition of the experimental diets (g kg$^{-1}$).

| | PSPP | PSPP0 | PSPP10 | PSPP20 | PSPP40 | PSPP80 |
|---|---|---|---|---|---|---|
| Fish meal | - | 150 | 150 | 150 | 150 | 150 |
| Corn meal | - | 200 | 200 | 200 | 200 | 200 |
| Soybean meal | - | 390 | 387 | 384 | 383 | 380 |
| Wheat flour | - | 70 | 70 | 70 | 70 | 70 |
| Rice bran | - | 150 | 150 | 150 | 135 | 100 |
| PSPP | - | 0 | 10 | 20 | 40 | 80 |
| Cellulose | - | 20 | 13 | 6 | 2 | 0 |
| Soybean oil | - | 5 | 5 | 5 | 5 | 5 |
| Premix | - | 10 | 10 | 10 | 10 | 10 |
| Vitamin C (98%) | - | 5 | 5 | 5 | 5 | 5 |
| Total | - | 1000 | 1000 | 1000 | 1000 | 1000 |
| Proximate composition of the experimental diets (%) | | | | | | |
| Crude protein | 10.1 | 32.80 | 35.2 | 34.5 | 33.6 | 32.3 |
| Crude lipid | 1.01 | 2.85 | 3.45 | 4.18 | 3.62 | 3.60 |
| Fiber | 25.73 | 3.68 | 4.36 | 5.45 | 5.21 | 6.44 |
| Ash | 8.02 | 7.59 | 8.38 | 7.96 | 7.87 | 7.99 |
| Dry matter | 95.13 | 99.16 | 97.05 | 96.16 | 98 | 97.69 |
| Gross Energy (cal/g) | 2731 | 4273 | 4278 | 4203 | 4185 | 4085 |

## 2.2. Culture Conditions

Three weeks before starting the experiment, floc inoculants were generated in each tank (150 L) by adding sea salt (400 g), dolomite (5 g), wheat flour (2 g), and molasses (5 g). After formation, the floc quantity was kept constant throughout the experiment at a level of approximately $8.21 \pm 0.15$ mL per tank, by maintaining the $NH_3$ concentration at $0.11 \pm 0.005$ mg L$^{-1}$ and modifying the C:N ratio (15:1) by adding molasses and probiotics (PondPlus, Bayer Thai Co., Ltd., Bangkok, Thailand) [53]. The C:N ratio was calculated using a diagrammatic representation of residual nitrogen grades and food intake [54].

## 2.3. Experimental Design

Nile tilapia fingerlings were purchased from a tilapia hatchery in Chiang Mai, Thailand. Fish were acclimatized and fed a commercial meal CP-9950 (Charoen Pokphand Foods Public Co., Ltd., Bangkok, Thailand) for two weeks. Prior to conducting additional experiments, twenty randomly selected fish were subjected to bacterial and parasite examinations to guarantee their health. A total of 300 Nile tilapia ($14.22 \pm 0.05$ g) were randomly assigned into five dietary treatment groups with PSPP supplemented as follows: control-PSPP0 (0 g kg$^{-1}$), PSPP10 (10 g kg$^{-1}$), PSPP20 (20 g kg$^{-1}$), PSPP40 (40 g kg$^{-1}$), and PSPP80 (80 g kg$^{-1}$). Fish were maintained in 150 L glass tanks. The experimental trials were conducted in triplicate with 20 fish per tank. Fish were fed twice daily at 8:30 a.m. and 4:30 p.m. at 3% body weight for eight weeks. Temperature, pH, and dissolved oxygen were maintained at 25–29 °C, 7.5–7.9, and 5 mg L$^{-1}$, respectively.

## 2.4. Growth Rates

Initial weight (IW), weight gain (WG), survival rate (SR), specific growth rate (SGR), and feed conversion rate (FCR) in Nile tilapia were determined after four and eight weeks of feeding trial as follows [55]:

$$\text{WG (g)} = \text{FW} - \text{IW}$$

$$\text{SGR}\left(\% \text{ day}^{-1}\right) = 100 \times \frac{\text{FW} - \text{IW}}{60 \text{ d}}$$

$$\text{FCR (g)} = \frac{\text{total feed given}}{\text{weight gain}}$$

$$\text{SR (\%)} = \frac{\text{final number}}{\text{initial number}} \times 100$$

Weights were measured using a balance with an accuracy of 0.1 g. Additionally, any dead fish were tallied and the mortality rate computed during the experiment.

### 2.5. Immune Response Analysis

#### 2.5.1. Sample Collection

To examine immunological activities, skin mucus and blood samples (6 fish/treatment) were collected. Before collecting samples, clove oils (5 mL L$^{-1}$) were used to anesthetize fish. For skin mucus sample collection, the experimental fish were gently rubbed for 2 min in a plastic bag containing 10 mL of 50 mM NaCl (Merck, Germany). The mixture was immediately transferred into new sterile tubes and centrifuged at $1500 \times g$ at 4 °C for 10 min. The mucus samples (1 mL) were then kept at −80 °C until used. For blood sampling, blood was collected according to the protocols previously reported [56]. Briefly, approximately 1 mL of fish blood was obtained using a 1 mL syringe at the caudal vein. Blood samples were promptly withdrawn and placed into sterile tubes (without adding anticoagulant). The samples were kept for an hour at room temperature and at 4 °C for a further hour. The samples were then centrifuged, and the serum samples were collected and stored at −80 °C for further analysis.

#### 2.5.2. Lysozyme and Peroxidase Assay

Lysozyme assays were conducted following the procedures reported by Parry, et al. [57], with the minor modifications in Van Doan et al. [55]. The equivalent unit of activity for each sample was calculated in accordance with the standard curve, which was constructed by plotting the decrease in the optical density value against the concentration, ranging from 0–20 μL mL$^{-1}$ of hen egg-white lysozyme (Sigma-Aldrich, Inc., USA) and represented as μg mL$^{-1}$ of serum.

Peroxidase activity was determined according to the protocols described in Quade and Roth [58] and Cordero, et al. [59], with minor modifications in Van Doan et al. [55]. The peroxidase assay was represented in units (U) per mg of skin mucus or serum proteins, where a unit was defined as the quantity of serum or mucus proteins that produced a change in absorbance equal to one.

### 2.6. Total RNA Extraction and Real-Time PCR (qPCR) Analysis

A total of 40–50 mg of fish tissues (gills and liver) was sampled for RNA extraction using TRIzol and an RNA extraction kit (Invitrogen, Waltham, MA, USA). The quality and quantity were measured using spectrophotometers (Thermo Fisher Scientific, Waltham, MA, USA) at wavelengths of 260 and 280 nm. The first-strand complementary DNA (cDNA) was synthesized with 1 μg total RNA using the iScript™ cDNA Synthesis Kit (BIO-RAD, Hercules, CA, USA). The analysis was conducted in triplicate using 100 ng of cDNA and iTaq Universal SYBR Green on a CFX Connect™ real-time PCR (BIO-RAD, Hercules, CA, USA). The qPCR was conducted at 95 °C for 30 s, 40 cycles of 95 °C for 15 s, and 60 °C for 30 s and followed by 95 °C for 15 s, 60 °C for 60 s, and 95 °C for 15 s. Expression levels was analyzed according to the $2^{-\Delta\Delta Ct}$ method [60] The qPCR results were normalized to the *18S rRNA* gene. The primers used for the qPCR analysis in this study are presented in Table 2.

**Table 2.** Primers used for the qPCR analysis in this study.

| Target Genes | Sequence (5′-3′) | Tm (°C) | Product Size (bp) | Ref. |
|---|---|---|---|---|
| *18S rRNA* | GTGCATGGCCGTTCTTAGTT CTCAATCTCGTGTGGCTGAA | 60 | 150 | [61] |
| *il-1* | GTCTGTCAAGGATAAGCGCTG ACTCTGGAGCTGGATGTTGA | 59 | 200 | [61] |
| *il-8* | CTGTGAAGGCATGGGTGTG GATCACTTTCTTCACCCAGGG | 59 | 196 | [61] |

**Table 2.** *Cont.*

| Target Genes | Sequence (5′-3′) | Tm (°C) | Product Size (bp) | Ref. |
|---|---|---|---|---|
| *lbp* | ACCAGAAACTGCGAGAAGGA GATTGGTGGTCGGAGGTTTG | 59 | 200 | [61] |
| *gst-α* | ACTGCACACTCATGGGAACA TTAAAAGCCAGCGGATTGAC | 60 | 190 | [61] |
| *gpx* | GGTGGATGTGAATGGAAAGG CTTGTAAGGTTCCCCGTCAG | 60 | 190 | [61] |
| *gsr* | CTGCACCAAAGAACTGCAAA CCAGAGAAGGCAGTCCACTC | 60 | 172 | [61] |

Note: F: Forward, R: Reverse, bp: base pair.

*2.7. Statistical Analysis*

The Kolmogorov–Smirnov test was used to evaluate the normality of the data. Means were compared using Duncan's multiple range test. Growth rates, immunological responses, and gene expression levels were analyzed using ANOVA analysis. SAS v9.4 statistical software (Cary, NC, USA) was used for all the statistical analyses [62]. $p < 0.05$ was denoted as a significant difference.

**3. Results**

*3.1. Growth Performance Analysis*

In this study, the growth parameters observed in different dietary treatment groups are presented in Table 3. There was no significant difference in final weight (FW), weight gain (WG), feed conversion ratio (FCR), or specific growth rate (SGR) between fish fed PSPP-supplemented diets and those fed only a basal diet (0 g kg$^{-1}$ PSPP) after four and eight weeks of the experimental trial. The survival rate (SR) of all treatment groups exceeded 95% at the conclusion of the feeding studies. No significant difference in any groups of the dietary PSPP-supplemented diets were detected ($p > 0.05$).

**Table 3.** Growth performances and feed utilization in Nile tilapia with different levels of PSPP-supplemented diets after four and eight weeks of the feeding trial. Different letters in the same row indicate statistically significant differences ($p < 0.05$). Data are presented as mean ± SE.

| | PSPP 0 | PSPP 10 | PSPP 20 | PSPP 40 | PSPP 80 |
|---|---|---|---|---|---|
| IW (g) | 14.22 ± 0.04 [a] | 14.23 ± 0.04 [a] | 14.25 ± 0.05 [a] | 14.15 ± 0.03 [a] | 14.23 ± 0.03 [a] |
| FW (g) | | | | | |
|     4 weeks | 28.60 ± 1.64 [a] | 29.05 ± 1.11 [a] | 28.69 ± 0.64 [a] | 29.73 ± 1.23 [a] | 29.67 ± 0.37 [a] |
|     8 weeks | 55.03 ± 0.64 [a] | 55.01 ± 0.91 [a] | 53.84 ± 1.35 [a] | 55.93 ± 1.36 [a] | 54.05 ± 0.79 [a] |
| SGR (%) | | | | | |
|     4 weeks | 2.32 ± 0.18 [a] | 2.37 ± 0.13 [a] | 2.33 ± 0.06 [a] | 2.47 ± 0.13 [a] | 2.45 ± 0.05 [a] |
|     8 weeks | 2.26 ± 0.02 [a] | 2.25 ± 0.03 [a] | 2.21 ± 0.04 [a] | 2.29 ± 0.04 [a] | 2.22 ± 0.03 [a] |
| WG (g) | | | | | |
|     4 weeks | 14.38 ± 1.61 [a] | 14.82 ± 1.13 [a] | 14.44 ± 0.59 [a] | 15.58 ± 1.21 [a] | 15.44 ± 0.39 [a] |
|     8 weeks | 40.81 ± 0.64 [a] | 40.78 ± 0.90 [a] | 39.59 ± 1.33 [a] | 41.78 ± 1.34 [a] | 39.82 ± 0.82 [a] |
| FCR | | | | | |
|     4 weeks | 1.00 ± 0.07 [a] | 1.03 ± 0.03 [a] | 1.11 ± 0.08 [a] | 0.98 ± 0.06 [a] | 1.01 ± 0.00 [a] |
|     8 weeks | 0.96 ± 0.03 [a] | 1.01 ± 0.03 [a] | 1.01 ± 0.02 [a] | 0.98 ± 0.02 [a] | 1.01 ± 0.02 [a] |
| SR (%) | | | | | |
|     4 weeks | 100 | 98 | 97 | 100 | 98 |
|     8 weeks | 98 | 95 | 95 | 98 | 98 |

Note: IW: Initial weight; FW: Final weight; SGR: Specific growth rate; WG: Weight gain; FCR: Feed conversion ratio; SR: Survival rate.

*3.2. Analysis of Skin Mucus Immune Responses*

Lysozyme and peroxidase activities in skin mucus in Nile tilapia after four and eight weeks of feeding are presented in Figure 1. No significant difference ($p > 0.05$) in lysozyme activity was observed between fish fed PSPP-supplemented diets and those fed only a basal diet (0 g kg$^{-1}$ PSPP) after four and eight weeks of feeding. Only fish fed the 20 g kg$^{-1}$

PSPP (PSPP20) diet had significantly ($p < 0.05$) higher peroxidase activity than those with other treatments after four weeks of feeding trial (Figure 1B). No significant difference was found in peroxidase activity after eight weeks of feeding ($p > 0.05$).

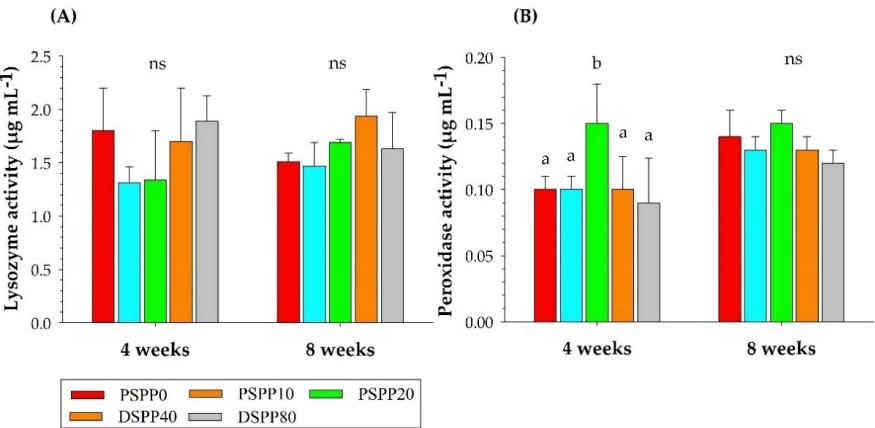

**Figure 1.** Lysozyme (**A**) and peroxidase (**B**) activity in the skin mucus of Nile tilapia after four and eight weeks of feeding with experimental diets: 0 g kg$^{-1}$ (PSPP0) control, 10 g kg$^{-1}$ (PSPP10), 20 g kg$^{-1}$ (PSPP20), 40 g kg$^{-1}$ (PSPP40), and 80 g kg$^{-1}$ (PSPP80) cultivated for eight weeks. Data are presented as mean $\pm$ SE. Significantly different levels are denoted by superscript letters ($p < 0.05$). "ns" denotes no significant difference.

### 3.3. Analysis of Serum Immune Responses

Serum immunological activities (lysozyme and peroxidase) were determined in this study using serum samples obtained after four and eight weeks of feeding. Figure 2 summarizes the impact of the experimental diets on serum immunological activity. Peroxidase and lysozyme activities in serum showed substantially higher levels in fish fed dietary PSPP-supplemented diets than those fed the basal diet without PSPP supplementation after four and eight weeks of the experimental trial. No statistically significant differences in lysozyme activity were detected in any of the dietary supplementation treatments ($p > 0.05$) after four or eight weeks of feeding. On the other hand, the PSPP10 diet substantially enhanced serum peroxidase activity compared to the control PSPP0 group ($p < 0.05$). At eight weeks post-feeding, a statistically significant change in the activity of peroxidase was detected between the PSPP-supplemented treatments (PSPP20 and PSPP80) and the PSPP0 control treatment ($p < 0.05$).

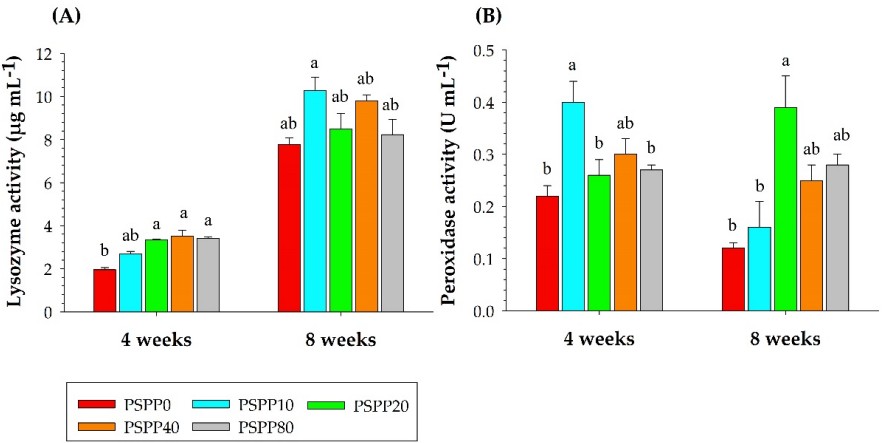

**Figure 2.** Lysozyme (**A**) and peroxidase (**B**) activity in the serum of Nile tilapia after four and eight weeks of feeding with experimental diets: 0 g kg$^{-1}$ (PSPP0) control, 10 g kg$^{-1}$ (PSPP10), 20 g kg$^{-1}$ (PSPP20), 40 g kg$^{-1}$ (PSPP40), and 80 g kg$^{-1}$ (PSPP80) cultivated for eight weeks. Data are presented as mean $\pm$ SE. Significantly different levels are denoted by superscript letters ($p < 0.05$).

### 3.4. Analysis of Immune and Antioxidant-Related Gene Expression

Fish tissues (gills and liver) were collected to investigate relative immune and antioxidant gene expressions by qPCR. Three relative immune genes (*lbp*, *il-1*, and *il-8*) and three antioxidant genes (*gsr*, *gst-α*, and *gpx*) were investigated in this study. A significant upregulation (approximately 2–2.7 fold) was found in the gill tissues of fish fed with dietary supplementation with PSPP compared to those fed the basal diet without PSPP supplementation (the control group, PSPP0). The greatest level of mRNA transcripts was observed in the PSPP20 diet groups (approximately 3.1–3.7 fold) in *lbp*, *gst-α*, and *gpx*, whereas *il-1* and *il-8* had the highest levels in the dietary PSPP40 and PSPP10 treatments, respectively (Figure 3).

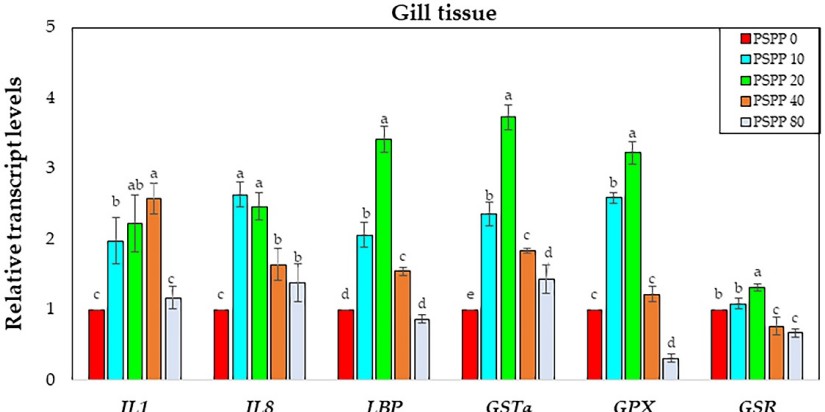

**Figure 3.** Expression transcript levels of interleukin-1 (*il-1*), interleukin-8 (*il-8*), lipopolysaccharide-binding protein (*lbp*), glutathione S-transferase-α (*GST-α*), glutathione peroxidase (*gpx*), and glutathione reductase (*gsr*) in gill tissues of Nile tilapia (*n* = 5) fed diets supplemented 0 g kg$^{-1}$ PSPP, 10 g kg$^{-1}$ PSPP, 20 g kg$^{-1}$ PSPP, 40 g kg$^{-1}$ PSPP, and 80 g kg$^{-1}$ PSPP after eight weeks of feeding. The mRNA transcript levels of immune and antioxidant genes were normalized by *18S rRNA*. The mRNA transcript level of the 0 g kg$^{-1}$ PSPP control group was set at 1. Data are presented as mean ± SE. Significantly different levels are denoted by superscript letters (*p* < 0.05).

A substantial difference in the mRNA transcripts of the examined genes was identified in liver tissues (Figure 4). Fish given the dietary PSPP20 expressed the greatest levels of mRNA transcripts in all target genes (except *lbp*) compared to the other dietary treatment and the control group (approximately 2–4.3 fold). *Lbp* expression was considerably upregulated in fish fed with dietary PSPP40 (approximately 2.5-fold), followed by PSPP20 (approximately 2.0-fold) and PSPP10 (approximately 1.8-fold).

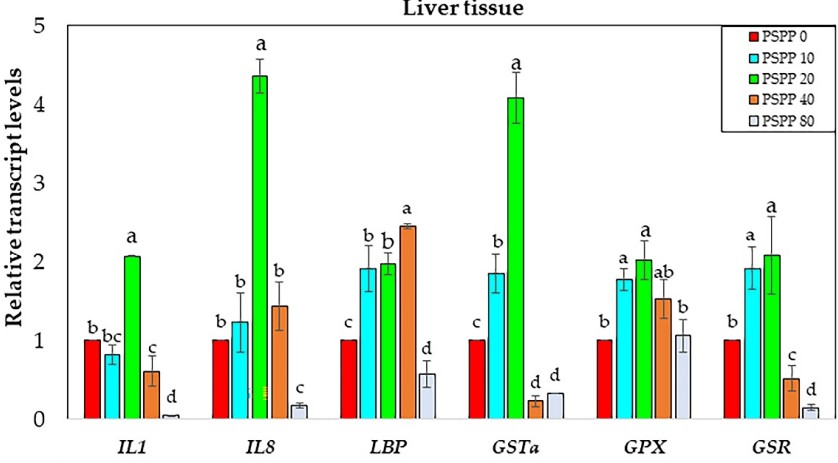

**Figure 4.** Expression transcript levels of interleukin-1 (*il-1*), interleukin-8 (*il-8*), lipopolysaccharide-binding protein (*lbp*), glutathione S-transferase-α (*gst-α*), glutathione peroxidase (*gpx*), and glutathione

reductase (*gsr*) in liver tissues of Nile tilapia ($n = 5$) fed diets supplemented 0 g kg$^{-1}$ PSPP, 10 g kg$^{-1}$ PSPP, 20 g kg$^{-1}$ PSPP, 40 g kg$^{-1}$ PSPP, and 80 g kg$^{-1}$ PSPP after eight weeks of feeding. The mRNA transcript levels of immune and antioxidant genes were normalized by *18S rRNA*. The mRNA transcript level of the 0 g kg$^{-1}$ PSPP control group was set at 1. Data are presented as mean $\pm$ SE. Significantly different levels are denoted by superscript letters ($p < 0.05$).

## 4. Discussion

The use of fruit by-products as feed additives in aquaculture is an efficient approach to conserve the environment and generate more income for farmers. By-products, such as peels and seeds, contain many substances with health-promoting effects [63–65].

After eight weeks of feeding, we observed that growth performance and feed consumption were unaffected by the PSPP supplement diets. The findings of this study were consistent with those of earlier studies on Jaraqui (*Semaprochilodus insignis*) and tambaqui fed passionfruit seed cake [66]; silver catfish (*Schilbe intermedius*) fed grape, orange, guava, and fig residues [67]; Nile tilapia and African catfish (*Clarias gariepinus*) fed dehydrated lemon peels [68]; Nile tilapia fed passionfruit seed oil and pomegranate peel [69,70]; rainbow trout (*Oncorhynchus mykiss*) fed dehydrate lemon peel [71]; Asian sea bass (*Lates calcarifer*) fed fermented lemon peel [72,73]; and giant freshwater prawn (*Macrobrachium rosenbergii*) fed biogas sludge meal [74], suggesting that the PSPP may not promote the production of digestive enzymes or intestinal absorption due to the large levels of soluble and insoluble fiber in PSPP [75]. Dietary fiber in PSPP has been shown by Vuolo, et al. [76] to decrease glucose and lipid absorption, resulting in less energy storage and increased lipid and glucose excretion. There was no discernible difference between the dietary PSPP-supplemented groups in this study. According to Ramli, et al. [33], PSPP has various valuable active components, including phenolic compounds, flavonoids, cyanogenic chemicals, anthocyanin, minerals, polysaccharides, and vitamins—which may account for its beneficial impact on growth performance [28,34,77]. Additionally, the PSPP contains considerable amounts of vitamin C that can be fortified into fish feed [78]. More investigations are needed to clarify the impact of these extracts on the growth performance of Nile tilapia cultivated in biofloc systems.

Skin mucus plays an important role in fish immune responses [79]. Lysozyme and peroxidase are important indicators of the immune defense system of fish; it has lytic action against bacteria and participates in phagocytic activity, neutrophil activation, and the complement system [80,81]. Lysozyme and peroxidase activities were greater in fish fed PSPP diets than in the control group after four and eight weeks of the feeding experiment. The addition of fruit by-products or extracts to diets, especially powdered passionfruit peel, has a beneficial effect on the immunological activity of Nile tilapia, striped catfish (*Pangasianodon hypophthalmus*), black rockfish (*Sebastes schlegelii*), and gilthead seabream (*Sparus aurata*) [61,82–84].

*il-1* and *il-8* are responsible for regulating inflammatory processes in the innate immune system to stimulate phagocytes to engulf microorganisms [85]. Antioxidant-related genes are involved in the glutathione protection mechanism, responsible for hydrogen peroxide removal ($H_2O_2$). A phase II xenobiotic metabolic enzyme, glutathione S-transferase (GST), combines with electrophilic chemicals to produce bigger endogenic complexes known as glutathione S-conjugates, which are then expelled from the body [86]. GPX transforms $H_2O_2$ into $H_2O$ via the oxidation of glutathione (GSH) to glutathione disulfide (GSSG). GSH is revived by GSR after it has been oxidized by the oxidative reduction of NADPH [87]. Increased immune responses and gene expression levels in fish fed powdered passionfruit peel are likely to be the result of an overall improvement in health and wellbeing due to a combination of several health benefits associated with dietary PSPP. These include (i) greater immunity against pathogens, indicated by elevated lysozyme and peroxidase levels in skin mucus and serum, and by elevated *il-1*, *il-8*, and *lbp* mRNA transcript levels

in the gills and liver tissues; (ii) enhanced antioxidant activity, indicated by elevated mRNA transcript levels of *gst-α, gpx* and *gsr*; and (iii) PSPP may stimulate the immune defense system in fish, thereby improving survival rates and disease resistance in fish.

The successful application of biofloc in aquaculture depends on the presence of both prebiotics and probiotics. The addition of PSPP in a biofloc aquaculture system may be involved in several processes, such as stimulating the proliferation of favorable bacteria, inhibiting the growth of pathogenic microorganisms, and improving the gastrointestinal condition of fish [61,88–90]. On the other hand, the recycling of nitrogen via its conversion to microbial biomass in biofloc increases the populations of favorable bacteria, enhancing host immunity [91].

## 5. Conclusions

Diets containing powdered passionfruit peel at concentrations of 10 to 20 g kg$^{-1}$ improved expression levels of innate immune and antioxidant-related genes in Nile tilapia cultured in a biofloc system. However, fish fed PSPP-supplemented feed had no significantly differences in growth performance; further studies should explore this issue to gain a better understanding of the impacts of PSPP in Nile tilapia.

**Author Contributions:** P.O.; investigation, N.V.L.; writing—original draft preparation, C.L.X.; formal analysis, S.W.; formal analysis, S.T.; resources, C.C.; data curation, N.M.; visualization, H.V.D.; conceptualization, project administration, and writing—review and editing. All authors have read and agreed to the published version of the manuscript.

**Funding:** This research work was partially supported by Chiang Mai University (CoE2565).

**Institutional Review Board Statement:** This study was conducted in accordance with international guidelines and approved by the Chiang Mai University Committee (No. RAGIACUC002/2565).

**Data Availability Statement:** The data presented in this study are available on request from the corresponding author.

**Acknowledgments:** The authors take this opportunity to thank the National Research Council of Thailand for supporting and helping with the study. This research work was partially supported by Chiang Mai University.

**Conflicts of Interest:** The authors declare no conflict of interest.

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
