# Peer review of "Passionfruit (Passiflora edulis) Peel Powder Stimulates the Immune and Antioxidant Defense System in Nile Tilapia, Oreochromis niloticus, Cultivated in a Biofloc System"

_fishes, doi:10.3390/fishes7050233_

Round 1

Reviewer 1 Report

Keywords can be improved by adding: antioxidant defensive system; feed additives

Line 65: please provide the Latin name for tambaqui

Please be more strict when mentioned name of fish as well other species. The use of "tilapia" and "Nile tilapia" in the text is not good without an explanation that "tilapia" means "Nile tilapia".

The paragraph started in line 69: Please explain what bio-floc system is, make it clear to those who are not familiar with this system.

Two versions, “bio-floc” and “biofloc”, are used in the manuscript. I suggest to use “biofloc” through the whole text, including the Title and Abstract.

Line 85: what oil was used and what amount of oil was added?

Lines 87-88: “The estimated components of collected agricultural by-product samples were determined according to the AOAC [50] instructions” – this is not clear, needs clarification

Line 110: what salt was used – please specify  

Subchapter “2.2 Preparation of floc water” should be rewritten. Now it is impossible to understand what is the floc water and how floc water was prepared.

Lines 118-119: phrase “…examinations of their physical structures, gills, and main organs” difficult to accept. It evokes a lot of questions: what physical structures and main organs were examined, and what were the criteria?

Lines 119-120: it should be noted that 5 groups were equal in number of fish

Lines 123-124: it is necessary to clarify the diet - how much food did the fish receive each time?

Subchapter “2.5.1 Sample collection”: what volume of mucus was collected for each sample? This is not about collecting serum samples, but blood – please, correct in this subchapter and in other parts of the manuscript.

Line 148: what kind of incubation do you mean?

Line 269: I suggest to remove “biological”

Lines 270-273: I am not sure that the repetition of the aim this study is needed in Discussion.

Line 278: residue of what kind of fruit do you mean?

Lines 287-289: Sentence “However, growth performance in a diet supplemented with the 40 g kg-1 PSPP (PSPP40) was slightly better in FW, SGR, and WG than the control and the other PSPP-supplemented diets” should be removed because the mentioned differences are non-significant.

Lines 303-304: please specify what tilapia and catfish do you mean?  Please provide the Latin names for black rockfish, and gilthead seabream.

Lines 304-306: the sentence “These results indicate that the capacity of aquatic animals to use feed additives differs by species, developmental stage, and dietary components [84,85]” is poorly related to the previous text and little understood.

Lines 315: the statement that “Enhanced growth and efficiency of feed conversion in fish fed powdered passion fruit peel…” looks controversial for the obtained results – please correct.

Lines 332-333: the statement “….diets containing powdered passion fruit peel at concentrations of 10 to 20 g kg-1 increased growth, improved feed utilization…” looks controversial for the obtained results – please correct.

Line 336: “additives for aquatic species, especially Nile tilapia” should be replaced by “additives for Nile tilapia”.

Author Response

RESPONSES TO COMMENTS BY REFREE#1

Response: Thank you very much for your kind supports. In the following sections, you will find our responses to each of your points and suggestions. We are grateful for the time and energy you expended on our behalf.

Comment 1: Keywords can be improved by adding: antioxidant defensive system; feed additives  

Response: Thank you for your comments. The manuscript has been amended to include this information (see Lines 29-30, Revision Manuscript)

Comment 2: Line 65: please provide the Latin name for tambaqui

Response: Thank you for your comments. The manuscript has been amended to include this information (see Line 66, Revision Manuscript)

Comment 3. Please be more strict when mentioned name of fish as well other species. The use of "tilapia" and "Nile tilapia" in the text is not good without an explanation that "tilapia" means "Nile tilapia".

Response: Thank you for your comments. Experimental fish used in this study and others have been carefully checked and corrected throughout the manuscript (see Revision manuscript).

Comment 4: The paragraph started in line 69: Please explain what bio-floc system is, make it clear to those who are not familiar with this system.

Response: Thank you for your comments. The manuscript has been amended to include this information to read “Bio-floc technology (BFT) is an alternative approach that mixes the aggregates of algae, protozoa, or bacteria with particulate organic substances to improve water quality, waste treatment, and disease prevention in intensive aquaculture systems”(see Lines 69-71, Revision Manuscript)

Comment 5: Two versions, “bio-floc” and “biofloc”, are used in the manuscript. I suggest to use “biofloc” through the whole text, including the Title and Abstract.

Response: The word ‘bio-floc’ has been amended to read ‘biofloc’ throughout the manuscript.

Comment 6: Line 85: what oil was used and what amount of oil was added?

Response: Thank you for comment. The soybean oil was used in this study, and the amount was provided in Table 1. The manuscript has been amended to include this information to read “… To prepare the dough, the feedstuffs were blended, then soybean oil and distilled water were added” (see Lines 86-87, Revision Manuscript)

Comment 7: Lines 87-88: “The estimated components of collected agricultural by-product samples were determined according to the AOAC [50] instructions” – this is not clear, needs clarification

Response: Thank you for comment. To avoid confusion, the manuscript has been revised to read “The proximate composition of diets was determined using the techniques recommended by AOAC [52], and crude fat content was measured using an extractable matter machine (SoxtecTM 8000, Denmark)” (see Lines 90-93, Revision Manuscript).

Comment 8: Line 110: what salt was used – please specify 

Response: Thank you for your comment. The manuscript has been amended to include this information (see Lines 114, Revision Manuscript)

Comment 9: Subchapter “2.2 Preparation of floc water” should be rewritten. Now it is impossible to understand what is the floc water and how floc water was prepared.

Response: Thank you for your comment. The manuscript has been revised to read:

2.3. Culture conditions

Three weeks before the feeding experiment, floc formation was initiated by adding 400 g of salt, 2 g of wheat flour, and 5 g of molasses and dolomite to each tank. After formation the floc quantity was kept constant throughout the experiment at a level of approximately 8.21 ± 0.15 mL per tank, by maintaining NH3 concentration at 0.11 ± 0.005 mg L−1 and modifying the ratio of C:N (15:1) [52] by adding molasses and probiotics (PondPlus, Bayer Thai Co., Ltd.). The C:N ratio was calculated using a diagrammatic representation of residual nitrogen grades and food intake [53]” (see Lines 113-122, Revision Manuscript)

Comment 10: Lines 118-119: phrase “…examinations of their physical structures, gills, and main organs” difficult to accept. It evokes a lot of questions: what physical structures and main organs were examined, and what were the criteria?

Response: Thank you for your comments. To make sure that fish were healthy before starting the experiment, fish were subjected to bacterial and parasite examinations. To avoid confusion, the manuscript has been now revised to read “Prior to conducting additional experiments, twenty randomly selected fish were subjected to bacterial and parasite examinations to guarantee their health” (see Lines 126-129, Revision Manuscript).  

Comment 11: Lines 119-120: it should be noted that 5 groups were equal in number of fish

Response: Thank you for your comment. In this study, a total of 300 Nile tilapia fingerlings were divided into 5 treatment groups and the experimental trial was conducted in triplicates with 20 fish per tank (see Line 133, Revision Manuscript).

Comment 12: Lines 123-124: it is necessary to clarify the diet - how much food did the fish receive each time?

Response: Thank you for your comment. The manuscript has been amended to include this information to read “Fish were fed twice daily at 8:30 a.m. and 4:30 p.m.  at 3% body weight for eight weeks” (see Line 134 Revision Manuscript).

Comment 13: Subchapter “2.5.1 Sample collection”: what volume of mucus was collected for each sample? This is not about collecting serum samples, but blood – please, correct in this subchapter and in other parts of the manuscript.

Response: Thank you for your comment. The manuscript has been amended to include the information (see Lines 149-160, Revision Manuscript).

Comment 14: Line 148: what kind of incubation do you mean?

Response: Thank you for your comment. To avoid confusion, the manuscript has been now revised to read “The samples were then centrifuged, and the serum samples were collected…” (see Line 160, Revision Manuscript).

Comment 15: Line 269: I suggest to remove “biological”

Response: Thank you for your comment. The ‘biological’ has been deleted.

Comment 16: Lines 270-273: I am not sure that the repetition of the aim this study is needed in Discussion.

Response: Thank you for your comment. We agree with the reviewer that the aims of study should be removed from discussion part.  

Comment 17: Line 278: residue of what kind of fruit do you mean?

Response: Thank you for your comment. To avoid confusion, the manuscript has been now revised to read “… silver catfish (Schilbe intermedius) fed grape, orange, guava, and fig residues [66]…”

Comment 18: Lines 287-289: Sentence “However, growth performance in a diet supplemented with the 40 g kg-1 PSPP (PSPP40) was slightly better in FW, SGR, and WG than the control and the other PSPP-supplemented diets” should be removed because the mentioned differences are non-significant.

Response: Thank you for your comment. The information has been removed.

Comment 19: Lines 303-304: please specify what tilapia and catfish do you mean?  Please provide the Latin names for black rockfish, and gilthead seabream.

Response: Thank you for your comment. To avoid confusion, the manuscript has been now revised to read “… the addition of fruit by-products or extracts to diets, especially powdered passion fruit peel, has a beneficial effect on the immunological activity of Nile tilapia, striped catfish (Pangasianodon hypophthalmus), black rockfish (Sebastes schlegelii), and gilthead seabream (Sparus aurata)…” (see Lines 315-317, Revision Manuscript).

Comment 20: Lines 304-306: the sentence “These results indicate that the capacity of aquatic animals to use feed additives differs by species, developmental stage, and dietary components [84,85]” is poorly related to the previous text and little understood.

Response: Thank you for your comment. The information has been removed.

Comment 21: Lines 315: the statement that “Enhanced growth and efficiency of feed conversion in fish fed powdered passion fruit peel…” looks controversial for the obtained results – please correct.

Response: Thank you for your comment. The manuscript has been now revised to read “Increased immune response and gene expression level in fish…” (see Lines 328-329, Revision Manuscript).

Comment 22: Lines 332-333: the statement “…. diets containing powdered passion fruit peel at concentrations of 10 to 20 g kg-1 increased growth, improved feed utilization…” looks controversial for the obtained results – please correct.

Response: Thank you for your comment. The manuscript has been now revised to read “Diets containing powdered passion fruit peel at concentrations of 10 to 20 g kg-1 improved expression levels of innate immune and antioxidant-related genes in Nile tilapia. However, fish fed PSPP-supplemented had no significantly difference on growth performance, further studies should explore this issue to gain better understanding the impacts of PSPP in Nile tilapia” (see Lines 345-350, Revision Manuscript).

Comment 23: Line 336: “additives for aquatic species, especially Nile tilapia” should be replaced by “additives for Nile tilapia”.

Response: Thank you for your comment. The manuscript has been now revised to read “These findings suggest that the addition of PSPP in basal diets might be employed as feed additives for Nile tilapia” (see Lines 348-349, Revision manuscript).

Reviewer 2 Report

The article is interesting and shows changes in the expression of the immune system and oxidative stress genes when using PSPP in diets for tilapia cultured in biofloc systems. I consider that the article provides adequate information on the use of passion fruit peel. However, the discussion should be complemented by the answers to the questions I ask the authors. The article could be accepted after major corrections.

Author Response

RESPONSES TO COMMENTS BY REFREE#2

Comment 1: The article is interesting and shows changes in the expression of the immune system and oxidative stress genes when using PSPP in diets for tilapia cultured in biofloc systems. I consider that the article provides adequate information on the use of passion fruit peel.

Response: Thank you very much for your kind supports. In the following sections, you will find our responses to each of your points and suggestions. We are grateful for the time and energy you expended on our behalf.

Comment 2: However, the discussion should be complemented by the answers to the questions I ask the authors. The article could be accepted after major corrections.

Response: Thank you for your comments. However, we could not find any comment of yours from the journal dashboard. I wonder if it would be possible for you to send us your comments for further corrections.

Reviewer 3 Report

The manuscript is suitable for publication, and presents relevant results but needs some adjustments.

Needs adjustments to Keywords that are highlighted in the text.

Needs adjustments to the conclusion that is highlighted in the text.

Author Response

RESPONSES TO COMMENTS BY REFREE#3

The manuscript is suitable for publication and presents relevant results but needs some adjustments. Needs adjustments to Keywords that are highlighted in the text. Needs adjustments to the conclusion that is highlighted in the text.

Response: Thank you very much for your kind words about our paper. In the following sections, you will find our responses to each of your points and suggestions. We are grateful for the time and energy you expended on our behalf.

Comment 1: Suggestion keywords: feed additives, immune gene expression, antioxidant gene expressions.  

Response: Thank you for your comments. The keywords have been revised to read “Feed additives; Immune gene expressions; Antioxidant defensive system (see Lines 29-30, Revision Manuscript)

Comment 2: Need to review the word “explosive”

Response: Thank you for your comment. The manuscript has been revised to read “… high growth, resistance to stress …” (see Lines 35, Revision Manuscript).

Comment 3: Line 182, “there was NO significant difference…”, need to review the conclusion about increased growth.

Response: Thank you for your comment. The manuscript has been revised to read “Diets containing powdered passion fruit peel at concentrations of 10 to 20 g kg-1 improved expression levels of innate immune and antioxidant-related genes in Nile tilapia. However, fish fed PSPP-supplemented had no significantly difference on growth performance, further studies should explore this issue to gain better understanding the impacts of PSPP in Nile tilapia” (see Lines 346-351, Revision Manuscript).

Comment 4: Line 294-295, growth performance needs more investigations… consider this in the conclusion.

Response: Thank you for your comment. The manuscript has been revised (see Lines 346-351, Revision Manuscript)

Round 2

Reviewer 2 Report

I have a few observations. Please check them.

Figure 2. Check peroxidase activity at 8 weeks, letters are wrongly assigned.

Genes abbreviations must be in lowercase and italics.

Author Response

RESPONSES TO COMMENTS BY REFREE#2, Review Report (Round 2)

I have a few observations. Please check them.

Response: Thank you very much for your kind supports. In the following sections, you will find our responses to each of your points and suggestions. We are grateful for the time and energy you expended on our behalf.

Comment 1: Figure 2. Check peroxidase activity at 8 weeks, letters are wrongly assigned.

Response: Thank you for improving this point. Statistical analysis has been carefully re-checked and letters have been corrected. 

Comment 2: Genes abbreviations must be in lowercase and italics.

Response: Thank you for your comment. The gene abbreviations have been amended to be in lowercase and italic throughout the manuscript.
